# LATENT CONVERGENT CROSS MAPPING

**Edward De Brouwer**[*†]
ESAT-STADIUS
KU LEUVEN
Leuven, 3001, Belgium
edward.debrouwer@esat.kuleuven.be

**Adam Arany**[*]
ESAT-STADIUS
KU LEUVEN
Leuven, 3001, Belgium
adam.arany@esat.kuleuven.be

**Jaak Simm**
ESAT-STADIUS
KU LEUVEN
Leuven, 3001, Belgium
jaak.simm@esat.kuleuven.be

**Yves Moreau**
ESAT-STADIUS
KU LEUVEN
Leuven, 3001, Belgium
moreau@esat.kuleuven.be

## ABSTRACT

Discovering causal structures of temporal processes is a major tool of scientific inquiry because it helps us better understand and explain the mechanisms driving a phenomenon of interest, thereby facilitating analysis, reasoning, and synthesis for such systems. However, accurately inferring causal structures within a phenomenon based on observational data only is still an open problem. Indeed, this type of data usually consists in short time series with missing or noisy values for which causal inference is increasingly difficult. In this work, we propose a method to uncover causal relations in chaotic dynamical systems from short, noisy and *sporadic* time series (that is, incomplete observations at infrequent and irregular intervals) where the classical convergent cross mapping (CCM) fails. Our method works by learning a Neural ODE latent process modeling the state-space dynamics of the time series and by checking the existence of a continuous map between the resulting processes. We provide theoretical analysis and show empirically that Latent-CCM can reliably uncover the true causal pattern, unlike traditional methods.

## 1 INTRODUCTION

Inferring a right causal model of a physical phenomenon is at the heart of scientific inquiry. It is fundamental to how we understand the world around us and to predict the impact of future interventions (Pearl, 2009). Correctly inferring causal pathways helps us reason about a physical system, anticipate its behavior in previously unseen conditions, design changes to achieve some objective, or synthesize new systems with desirable behaviors. As an example, in medicine, causality inference could allow predicting whether a drug will be effective for a specific patient, or in climatology, to assess human activity as a causal factor in climate change. Causal mechanisms are best uncovered by making use of interventions because this framework leads to an intuitive and robust notion of causality. However, there is a significant need to identify causal dependencies when only observational data is available, because such data is more readily available as it is more practical and less costly to collect (*e.g.*, relying on observational studies when interventional clinical trials are not yet available).

However, real-world data arising from less controlled environment than, for instance, clinical trials poses many challenges for analysis. Confounding and selection bias come into play, which bias standard statistical estimators. If no intervention is possible, some causal configurations cannot be identified. Importantly, with real-world data comes the major issue of missing values. In particular, when collecting longitudinal data, the resulting time series are often *sporadic*: sampling is *irregular*

---

[*]Both authors contributed equally
[†]Corresponding author

in time and across dimensions leading to varying time intervals between observations of a given variable and typically multiple missing observations at any given time. This problem is ubiquitous in various fields, such as healthcare (De Brouwer et al., 2019), climate science (Thomson, 1990), or astronomy (Cuevas-Tello et al., 2010).

A key problem in causal inference is to assess whether one temporal variable is causing another or is merely correlated with it. From assessing causal pathways for neural activity (Roebroeck et al., 2005) to ecology (Sugihara et al., 2012) or healthcare, it is a necessary step to unravel underlying generating mechanisms. A common way to infer causal direction between two temporal variables is to use Granger causality (Granger, 1969), which defines "predictive causality" in terms of the predictability of one time series from the other. A key requirement of Granger causality is then separability (i.e., that information about causes are not contained in the caused variable itself). This assumption holds in purely stochastic linear systems, but fails in more general cases (such as weakly coupled nonlinear dynamical systems) (Sugihara et al., 2012). To address this nonseparability issue, Sugihara *et al.* (Sugihara et al., 2012) introduced the Convergent Cross Mapping (CCM) method, which is based on the theory of chaotic dynamical systems, particularly on Takens' theorem. This method has been applied successfully in various fields such as ecology, climatology (Wang et al., 2018), and neuroscience (Schiecke et al., 2015). However, as the method relies on embedding the time series under study with time lags, it is highly sensitive to missing values and usually requires long uninterrupted time series. This method is thus not applicable in settings with repeated short sporadic time series, despite their occurrence in many practical situations.

To address this important limitation, we propose to learn the causal dependencies between time series by checking the existence of convergent cross mappings between latent processes of those time series. Using a joint model across all segments of sporadically observed time series and forcing the model to learn the inherent dynamic of the data, we show that our method can detect causal relationship from short and sporadic time series, without computing delay embeddings. To learn a continuous time latent representation of the system's state-space, we leverage GRU-ODE-Bayes (De Brouwer et al., 2019), a recently introduced filtering method that extends the Neural ODE model (Chen et al., 2018). Importantly for causal inference, the filtering nature of the model makes sure no future information can leak into the past. We then check the existence of continuous maps between the learnt latent representations and infer the causal direction accordingly.

In a series of increasingly challenging test cases, our method accurately detects the correct causal dependencies with high confidence, even when fed very few observations, and outperforms competing methods such as multi-spatial CCM or CCM with multivariate Gaussian process interpolation.

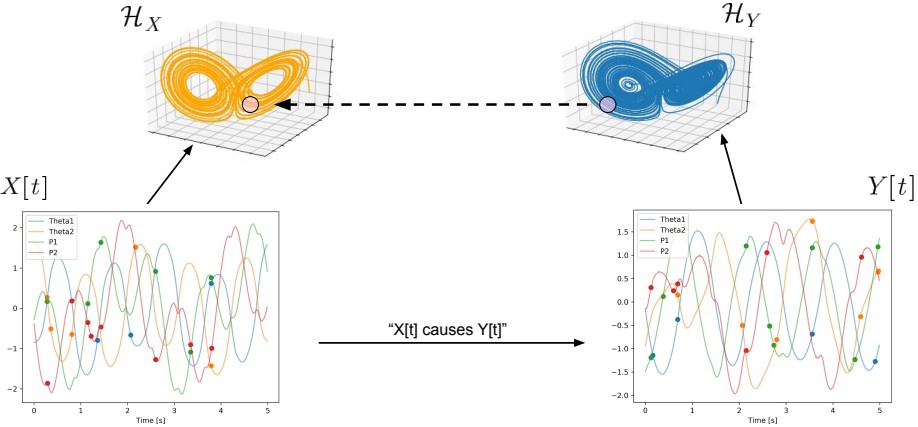

Figure 1: Schematic of the Latent-CCM rationale. If $X[t]$ causes $Y[t]$, there exists a continuous map (dotted line) from the latent process of $Y$ ($\mathcal{H}_Y$) to the latent process of $X$ ($\mathcal{H}_X$).

## 2 RELATED WORK

**CCM to address failure of Granger causality.** Granger causality (Granger, 1969) provided the first significant framework to infer causal dependencies from time series. Relying on predictability between dynamical systems, it was extended to account for different limitations, such as nonlinearity (Chen et al., 2004) or instantaneous relationships (Schiatti et al., 2015). However, the assumption of separability of information between causative and caused variables leads to the failure of the Granger paradigm for a significant number of time series coupling scenarios (Sugihara et al., 2012) (see Appendix D for a revealing worked out example). Convergent Cross Mapping, a technique based on nonlinear state space reconstruction was introduced to tackle this issue (Sugihara et al., 2012). Recently, several works have proposed extensions of CCM, such as the extended CCM, to address issues such as synchrony (Ye et al., 2015) or to improve the discrimination of the confounding case (Benkő et al., 2018). Synchrony occurs when one time series can be expressed as a function of the other (*e.g.* $Y(t) = \phi(X(t))$ and attractors of both dynamical systems become homeomorphic to each other (Rulkov et al., 1995). This occurs when coupling between two chaotic system is too strong. Confounding, on the other hand, occurs when two variables are causally driven by a third one. In general we say that $X$ confounds the relation between $Y$ and $Z$ if $X$ causes both $Y$ and $Z$.

Huang et al. (2020) also proposed to predict directly the driving time series from the driven one with reservoir computing, bypassing the delay embedding step, making it more robust to noise. However, those methods still require long regularly sampled time series.

**Causality for short or sporadic time series.** Short time series are very common in practice and there has been some work proposing to learn causality from short time series relying on state space reconstruction. Ma et al. (2014) proposed a method for short, fully observed, unique time series. Multi-spatial CCM (Clark et al., 2015), considered the problem of inferring causality from several short fully observed snippets of the same dynamical system by computing delay embeddings compatible with the lengths of the time series and aggregating them. In comparison, on top of addressing irregular sampling, our approach computes more informative state-space representations by sharing a model across all segments. Techniques to infer causal direction from incomplete time series have also been proposed, but all are relying on the Granger causality framework, which limits their applicability to separable dynamical systems. They use direct partial correlations on regularly sampled data (but with missing values) (Elsegai, 2019) or generalizations of similarity measures for sporadic time series (Bahadori & Liu, 2012). To the best of our knowledge, this is the first work investigating the identification of causal dependencies from short sporadic time series using state-space reconstruction.

## 3 METHOD

We consider the problem of inferring a causal dependency between two temporal variables from several segments of their multivariate time series $X[t] \in \mathbb{R}^{d_X}$ and $Y[t] \in \mathbb{R}^{d_Y}$. We assume that $X[t]$ and $Y[t]$ have been generated by an unknown dynamical system. In this work, we refer to the dynamical system of a time varying variable $X$ as the smallest dynamical system that fully describes the dynamics of $X$. As an example, let's consider the following system of ODEs representing the dynamics of $X$ and $Y$:

$$\frac{dX(t)}{dt} = f(X(t)) \tag{1}$$

$$\frac{dY(t)}{dt} = g(X(t)) + h(Y(t)). \tag{2}$$

The dynamical system of $X$ is given by Equation (1). On the other hand, the dynamical system of $Y$ is Equation (1) + (2) as Equation (1) is required to describe the dynamics of $Y$.

To account for the more general and most frequent case, we consider those time series are only observed in segments of finite duration. $X[t]$ and $Y[t]$ then consist of collections of $N$ short time series $(X^1[t],...,X^N[t])$ and $(Y^1[t],...,Y^N[t])$ respectively. Importantly, each segment of $X$ and $Y$ is observed concomitantly. To proceed with a lighter notation, we'll drop the superscript when referring to a segment of time series.

Each of those time series is also *sporadic* namely the are not regularly sampled and not all dimensions are observed each time.

In this work, we define the notion of causality by considering the equations of the dynamical system as a structural causal model. In this framework, X causes Y if $p(Y|do(X)) \neq P(Y)$ where $do(X)$ is an intervention on $X$ (Pearl, 2009). Then, if $X$ causes $Y$, $X$ is part of the dynamical system of $Y$ ($X$ is required to describe the dynamics of $Y$). In the case of the example described by Equations 1 and 2, $X$ causes $Y$ if $g(\cdot)$ is not a constant function.

## 3.1 CONVERGENT CROSS MAPPING AND TAKENS' THEOREM

CCM aims at discovering the causal direction between temporal variables in dynamical systems by checking if the state-space dynamics of their time series can be recovered from one another. As shown above, if $X$ causes $Y$, $X$ is then contained in the dynamical system of $Y$ and it should be possible to recover a representation of the dynamical system of $X$ from the dynamical system of $Y$.

A common way to obtain a representation of a dynamical system from its time series relies on Takens' embedding theorem (Takens, 1981).

Let $X[t] \in \mathbb{R}^{d_X}$ be issued from a chaotic dynamical system that has a strange attractor $\mathcal{M}$ with box-counting dimension $d_M$, where we define an attractor as the manifold toward which the state of a chaotic dynamical system tends to evolve. The dynamics of this system are specified by a flow on $\mathcal{M}$, $\phi_{(\cdot)}(\cdot) : \mathbb{R} \times \mathcal{M} \to \mathcal{M}$, where $\phi_\tau(\mathcal{M}_t) = \mathcal{M}_{t+\tau}$ and $\mathcal{M}_t$ stands for the point on the manifold at time index $t$. This flow is encoded in the ODE of the system. The observed time series $X[t]$ is then obtained through an observation function $f_{obs}(\cdot) : X[t] = f_{obs}(\mathcal{M}_t)$. Takens' theorem then states that a delay embedding $\Phi$ with delay $\tau$ and embedding dimension $k$

$$\Phi_{\phi,\alpha}^{k,\tau}(\mathcal{M}_t) = (\alpha(\phi_0(\mathcal{M}_t)), \alpha(\phi_{-\tau}(\mathcal{M}_t)), \ldots, \alpha(\phi_{-k\tau}(\mathcal{M}_t)))$$

is an embedding of the strange attractor $\mathcal{M}$ if $k > 2d_M$ and $\alpha : \mathbb{R}^{d_M} \to \mathbb{R}$ is a twice-differentiable observation function. More specifically, the embedding map $\Phi$ is a diffeomorphism between the original strange attractor manifold $\mathcal{M}$ and a shadow attractor manifold $\mathcal{M}'$ generated by the delay embeddings. Under these assumptions, one can then theoretically reconstruct the original time series from the delay embedding.

The simplest observation function $\alpha$ consists in simply taking one of the dimensions of observations of the dynamical system. In this case, writing $X_i[t]$ as the $i$-th dimension of $X[t]$, Takens' theorem ensures that there is a diffeomorphism between the original attractor manifold of the full dynamical system and the shadow manifold $\mathcal{M}'$ that would be generated by $X'[t] = (X_i[t], X_i[t-\tau], \ldots, X_i[t-k\tau])$. To see how this theorem can be used to infer the causal direction, let us consider the manifold $\mathcal{M}_Z$ of the joint dynamical system resulting of the concatenation of $X[t]$ and $Y[t]$. We then generate two shadow manifolds $\mathcal{M}'_X$ and $\mathcal{M}'_Y$ from the delay embeddings $X'[t] = (X_i[t], X_i[t-\tau], \ldots, X_i[t-k\tau])$ and $Y'[t] : (Y_j[t], Y_j[t-\tau], \ldots, Y_j[t-k\tau])$. Now, if X unidirectionally causes Y (*i.e.*, Y does not cause X), it means that X is part of an autonomous dynamical system and that Y is part of a larger one, containing X. The attractor of Y is then the same as the one of the joint dynamical system $Z$. By contrast, the attractor of X is only a subset of it. From Taken's theorem, it is theoretically possible to recover the original $\mathcal{M}_Z$ from $\mathcal{M}'_Y$ and hence, by extension, recover $\mathcal{M}'_X$ from $\mathcal{M}'_Y$. However, the contrary is not true and it is in general not possible to recover $\mathcal{M}'_Y$ from $\mathcal{M}'_X$.

The CCM algorithm uses this property to infer causal dependency. It embeds both dynamical systems $X$ and $Y$ and use $k$-nearest neighbors to predict points on $\mathcal{M}'_X$ from $\mathcal{M}'_Y$ and inversely. The result then consists in the correlation of the predictions with the true values. We write $\mathbb{Ccm}(X, Y)$ the Pearson correlation for the task of reconstructing $\mathcal{M}'_X$ from $\mathcal{M}'_Y$,

$$\mathbb{Ccm}(X, Y) = \mathbb{Corr}(\mathcal{M}'_X, \hat{\mathcal{M}}'_X)$$

where $\hat{\mathcal{M}}'_X$ stands fro the prediction of $\mathcal{M}'_X$ obtained from $\mathcal{M}'_Y$. Importantly, this measure is non-symmetric as an non-injective map between $\mathcal{M}'_X$ and $\mathcal{M}'_Y$ would lead to an accurate reconstruction being possible in one direction only.

To infer that there is a causal link between the predictor dynamical system and the predicted one, this correlation should be high and, importantly, increase with the length of the observed time series, as the observed manifolds become denser.

The potential results are then interpreted in the following way (1) $X$ causes $Y$ if one can reconstruct with high accuracy $\mathcal{M}'_X$ from $\mathcal{M}'_Y$; (2) $X$ and $Y$ are not causally related (but not necessarily statistically independent) if nor $\mathcal{M}'_X$ nor $\mathcal{M}'_Y$ can be reconstructed from the other; (3) $X$ and $Y$ are in a circular causal relation if both $\mathcal{M}'_Y$ and $\mathcal{M}'_X$ can be reconstructed from the other. In the extreme case of strong coupling, the two systems are said to be in synchrony, and it becomes hard to distinguish between unidirectional or bidirectional coupling (Ye et al., 2015).

## 3.2 NEURAL ODEs

Many continuous-time deterministic dynamical systems are usefully described as ODEs. But in general, not all dimensions of the dynamical system will be observed so that the system is better described as an ODE on a continuous latent process $H(t)$, conditioned on which the observations $X[t]$ are generated. For instance, when observing only one dimension of a 2 dimensional dynamical system, we cannot find a flow $\phi_t(X)$ on that single dimension variable, but we can find one on the latent process $H(t)$. We then have the following description of the dynamics:

$$X[t] = g(H[t]) \;\; \text{with} \;\; \frac{dH(t)}{dt} = f_\theta(H(t), t) \tag{3}$$

where $\theta$ represents the parameters of the ODE, $f_\theta(\cdot)$ is a uniformly Lipschitz continuous function and $g(\cdot)$ is a continuous function. Learning the dynamics of the system then consists in learning those parameters $\theta$ from a finite set of (potentially noisy) observations of the process $X$. Neural ODEs (Chen et al., 2018) parametrize this function by a neural network. Learning the weights of this network can be done using the adjoint method or by simply backpropagating through the numerical integrator. Note that one usually allows X[t] to be stochastic (*e.g.* observation noise). In that case, the mean of $X[t]$ (rather than $X[t]$ itself) follows Equation 3.

## 3.3 CAUSAL INFERENCE WITH LATENT CCM

A key step in the CCM methodology is to compute the delay embedding of both time series: $\Phi(X[t])$ and $\Phi(Y[t])$. However, when the data is only sporadically observed at irregular intervals, the probability of observing the delayed samples $X_i[t], X_i[t - \tau], \ldots, X_i[t - k\tau]$ is vanishing for any $t$. $X'[t]$ and $Y'[t]$ are then never fully observed (in fact, only one dimension is observed) and nearest neighbor prediction cannot be performed. What is more, short time series usually do not allow to compute a delay embedding of sufficient dimension ($k$) and lag ($\tau$) (Clark et al., 2015).

Instead of computing delay embeddings, we learn the dynamics of the process with a continuous-time hidden process parametrized by a Neural ODE (as in Eq. 3) and use this hidden representation as a complete representation of the state-space, therefore eliminating the need for delay-embedding that was limiting the applicability of CCM to long, constant sampling time series. A graphical representation of the method is shown on Figure 1.

To infer causality between temporal variables from their time series $X[t]$ and $Y[t]$, the first step is to train two GRU-ODE-Bayes models (De Brouwer et al., 2019), a filtering technique that extends Neural-ODEs. Being a filtering approach, GRU-ODE-Bayes ensures no leakage of future information backward in time, an important requirement for our notion of causality. The continuity of the latent process is also important as it provides more coverage of the attractor of the dynamical system. Indeed, a constant latent process in between observations (such as obtained with a classical recurrent neural network such as GRU) would lead to fewer unique latent process observations.

The same model is used for all segments of each time series and is trained to minimize forecasting error. We learn the observation function $g$, the ODE $f_\theta$ and the continuous-time latent process $H(t)$. We write the resulting space of latent vectors from time series $X$ on all segments as $\mathcal{H}_X$.

Causality is then inferred by checking the existence of a continuous map between $\mathcal{H}_X$ and $\mathcal{H}_Y$. Analogously to CCM, we consider $X$ causes $Y$ if there exists a continuous map between $\mathcal{H}_Y$ and

$\mathcal{H}_X$. This is consistent because $\mathcal{H}_X$, just as the delay embedding $\Phi_X$, is a embedding of the strange attractor of the dynamical system as stated in Lemma 3.1 for which we give the proof in the Appendix E.

**Lemma 3.1.** *For a sporadic time series $X[t] \in \mathcal{X}$ satisfying the following dynamics,*

$$X[t] = g(H(t)) \; with \; \frac{dH(t)}{dt} = f_\theta(H(t), t)$$

*with $g(\cdot)$ and $f_\theta$ continuous functions. If there exists one observation function $\alpha_H \in \mathcal{C}^2 : \mathcal{X} \to \mathbb{R}$ along with a valid couple $(k, \tau)$ (in the Takens' embedding theorem sense) such that the map $\Phi^{k,\tau}_{g(\phi_H), \alpha_H}(H(t))$ is injective, the latent process $H(t)$ is an embedding of the strange attractor of the full dynamical system containing $X$.*

The requirement of $\Phi^{k,\tau}_{g(\phi_H), \alpha_H}$ being injective is not enforced in our architecture. However, with sufficient regularization of the network, it is satisfied in practice as shown by our results in Section 4.5.

The same reasoning as in CCM then applies to the latent process and causal direction can be inferred. The existence of a continuous map between the latent spaces of both time series is quantitatively assessed with the correlation between the true latents of the driven time series and the reconstructions obtained with a $k$-nearest neighbors model on the latents of the driven time series. For instance, for a direction $X \to Y$, we report the correlation between predictions of $\mathcal{H}_X$ obtained from $\mathcal{H}_Y$ and the actual ones ($\mathcal{H}_X$). A strong positive correlation suggests an accurate reconstruction and thus a causal link in the studied direction between the variables (*e.g.*, $X \to Y$). By contrast, a weak correlation suggests no causal link in that direction.

## 4 EXPERIMENTS

We evaluate the performance of our approach on data sets from physical and neurophysiology models, namely a double pendulum and neurons activity data. We show that our method detects the right causal topology in all cases, outperforming multi-spatial CCM, as well as baselines designed to address the sporadicity of the time series. The code is available at `https://github.com/edebrouwer/latentCCM`.

### 4.1 BASELINE METHODS

To the best of our knowledge, this is the first time CCM is applied to short sporadic time series. Indeed, because of missing variables, many standard approaches are simply not applicable. The main baseline consists in multi-spatial CCM (Clark et al., 2015) applied to regularly sampled data with a sampling rate similar to the one of the sporadic data. We also compare our approach to variants where multi-spatial CCM is applied to an interpolation of the sporadic time series using (1) linear interpolation and (2) univariate and multivariate Gaussian Processes (GP and MVGP). For the Gaussian Process, we chose a mixture of RBF and identity kernel and learn the parameters from the data. To model multivariate GPs (MVGP), we used the combination of a Matern and a periodic Matern kernel for the time dimension and used co-regionalization (Bonilla et al., 2008) with a full-rank interaction matrix. We then use the mean of the posterior process as the reconstruction subsequentially fed to the classical CCM method. Implementation was done with GPflow (Matthews et al., 2017).

We also compared our approach to non-CCM causal discovery methods such as PCMCI (Runge et al., 2019) and VARLinGAM (Hyvärinen et al., 2010). PCMCI uses conditional independence testing between time series at different lags to infer causal dependencies. VARLinGAM learns a graphical model of the longitudinal variables and their time lags, using the LinGAM method (Shimizu et al., 2006). These methods do not allow for short sporadic time series as input but a comparison with a less challenging non-sporadic variant of our datasets is presented in Appendix F.

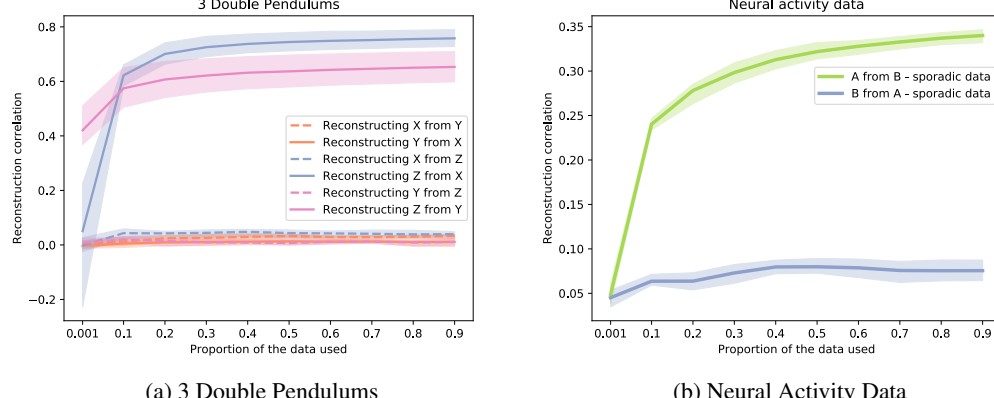

(a) 3 Double Pendulums               (b) Neural Activity Data

Figure 2: Pearson correlation of the reconstructions in function of the proportion of data used for the Latent-CCM. Solid line represent the mean over the 5 repeats and the shaded area the range of values. For both experiments, only the true causal directions show a convergent reconstruction. $\mathbb{C}cm_{full}$ is the score when the whole data is used. $\mathbb{C}cm_0$ is the score when only 100 samples of the latent space are used.

## 4.2 PERFORMANCE METRICS

Our method assesses causality by detecting convergent reconstruction accuracy between the latent processes of different time series. To account for both aspects in a single score, we use the difference between the correlation of the reconstruction and the target latent vector using the whole data ($\mathbb{C}cm_{full}$) and the correlation using only 100 sample points ($\mathbb{C}cm_0$), as shown on Figure 2 and suggested in Clark et al. (2015). The score for the causal coupling from $X[t]$ to $Y[t]$ is then defined as

$$\mathcal{S}c_{X \to Y} = \mathbb{C}cm_{full}(X, Y) - \mathbb{C}cm_0(X, Y).$$

with a higher score implying more confidence in a causal relationship. Additionally, to quantify the certainty about the presence of a causal edge in the data generation graph, we compare the obtained scores with the ones that would be obtained with CCM on fully observed but independent time series. We compute the Mann-Whitney $U$-statistics (Ma et al., 2014) and provide the corresponding $p$-value.

Nevertheless, in practice, one might not have access to the score of independent time series, making it difficult to assess from the score only if a causal relationship is present. To address this issue, we visualize the results graphically as shown in Figure 2. Causal directions should then stand out clearly and have the characteristic convergent pattern (Sugihara et al., 2012).

## 4.3 DOUBLE PENDULUM

**Description.** The double pendulum is a simple physical system that is chaotic and exhibits rich dynamical behavior. It consists of two point masses $m_1$ and $m_2$ connected to a pivot point and to each other by weightless rods of length $l_1$ and $l_2$, as shown on Figure 4 in Appendix A. The trajectories of the double pendulum are described by the time series of the angles of the rods with respect to the vertical ($\theta_1$ and $\theta_2$), as well as the angular momenta $p_1$ and $p_2$ conjugate to these angles. Each trajectory is then a collection of 4-dimensional vector observations.

To introduce causal dependencies from pendulum $X$ to $Y$, we include a non-physical asymmetrical coupling term in the update of the momentum conjugate to the first angle:

$$\dot{p}_1^Y = -\frac{\partial H^Y}{\partial \theta_1^Y} - 2 \cdot c_{X,Y}(\theta_1^Y - \theta_1^X),$$

where $c_{X,Y}$ is a coupling parameter. The term corresponding to a quadratic potential incorporated to the Hamiltonian of system $Y$ results in an attraction on system $Y$ by system $X$. Depending on the values of $c_{X,Y}$ and $c_{Y,X}$, we have different causal relationships between $X$ and $Y$. Namely, (1) $X$ causes $Y$ iff $c_{X,Y} \neq 0$, (2) $Y$ causes $X$ iff $c_{Y,X} \neq 0$ and (3) $X$ is not causally related to $Y$ if $c_{X,Y} = c_{Y,X} = 0$.

**Data generation**. We consider two cases of generating models. The first one consists of two double pendulums ($X[t]$ and $Y[t]$) with high observation noise with $Y$ causing $X$. In this case, we set $c_{X,Y} = 0$ and $c_{Y,X} = 0.2$. The second consists of 3 double pendulums ($X[t]$, $Y[t]$ and $Z[t]$), with one of them causing the other two ($c_{Z,X} = 0.5$, $c_{Z,Y} = 1$). We then infer the causal model relations between those 3 variables in a pairwise fashion (*i.e.* we infer the causal direction between all pairs of variables in the system). Remarkably, X and Y are here correlated but not causally related. Graphical representations of both considered cases are presented in Figure 3. Parameters of the pendulums (lengths and masses) are presented in Appendix A. We generate 5 trajectories with different initial conditions ($\theta_1 \sim \mathcal{N}(-1, 0.05)$ and $\theta_2 \sim \mathcal{N}(0.5, 0.05)$). We simulate observation noise by adding a random Gaussian noise $n$ to the samples with $n \sim \mathcal{N}(\mu = 0, \sigma = 0.1)$ for the first case and $n \sim \mathcal{N}(\mu = 0, \sigma = 0.01)$ for the second. To account for the short length of time series usually encountered in the real world, we randomly split the trajectories in windows of 10 seconds. To simulate sporadicity, we sample observation uniformly at random with an average rate of 4 samples per second. Furthermore, for each of those samples, we apply an observation mask that keeps each individual dimension with probability 0.3. This whole procedure leads to a sporadic pattern as shown in Figure 5 of Appendix A. We used 80% of available windows for training and used the remaining 20% for hyperparameter tuning with the MSE prediction on future samples used as model selection criterion. More details on this procedure is given in Appendix G.

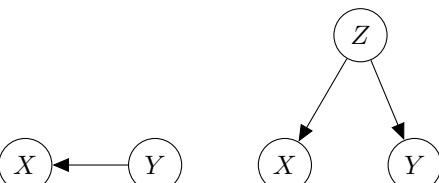

Figure 3: Graphical model representation of both cases considered in the main body of the paper. Left: Case 1. Right: Case 2 (confounding).

## 4.4 NEURAL ACTIVITY DATA

We also evaluate our approach on neural activity data. We generate time series of the average membrane potential of two populations of leaky integrate-and-fire neurons with alpha-function shaped synaptic currents (*iaf_psc_alpha*) simulated by NEST-2.20.0 (Fardet et al., 2020). Each neuron population contains 100 units with sparse random excitatory synapses within the population. We consider two cases, one where population A unidirectionally excites population B and another case where both populations fire independently. To account for the short and sporadic nature of real-world data, we generate 5,000 windows of 20 seconds from which we sample 1 observation every second on average. This leads to 20 samples being available on average per time window.

## 4.5 RESULTS

Results over 5 repeats for the double pendulums and the neural activity data are presented in Table 1 and in Figure 2. Because this method uses different metrics for inferring the underlying causal graph, the results for PCMCI are presented in Appendix F, where we show that the method cannot reliably infer the generative causal dependencies in our data.

Table 1: Average reconstruction scores $\mathcal{S}_c$ (and their standard deviations) in all directions for the double pendulum and neural activity experiments. Standard deviations are computed using 5 repetitions. Significant correlations compared to noncoupled dynamical systems are in bold ($p < 0.01$). Significance is computed using the Mann-Whitney rank test. Our approach detects the correct causal structure. ✓and ✗ highlight correct and wrong direction detection respectively.

| DATA | MULTI-SPATIAL CCM | LINEAR | MVGP | LATENT-CCM |
|---|---|---|---|---|
| PENDULUM | | | | |
| CASE 1 | | | | |
| $X \rightarrow Y$ | -0.017 ± 0.037 | 0.001 ± 0.006 | 0.009 ± 0.014 | 0.001 ± 0.013 |
| $\mathbf{X \leftarrow Y}$ | 0.018 ± 0.0056 | -0.011 ± 0.01 | -0.001 ± 0.021 | 0.055 ± 0.001 |
| $AUC_{1 \rightarrow 2}$ | 0.2 (P=0.98)✓ | 0.49 (P=0.52) ✓ | 0.66 (P=0.11) ✓ | 0.55 (P=0.35) ✓ |
| $AUC_{2 \rightarrow 1}$ | 0.6 (P=0.23)✗ | 0.17 (P=0.99) ✗ | 0.44 (P=0.67) ✗ | **1 (P<0.001) ✓** |
| CASE 2 | | | | |
| $X \rightarrow Y$ | 0.488 ± 0.074 | -0.006 ± 0.005 | -0.005 ± 0.009 | 0.001 ± 0.005 |
| $X \leftarrow Y$ | 0.181 ± 0.119 | -0.01 ± 0.01 | -0.007 ± 0.012 | 0.009 ± 0.014 |
| $X \rightarrow Z$ | 0.054 ± 0.021 | -0.002 ± 0.007 | -0.002 ± 0.014 | 0.019 ± 0.017 |
| $\mathbf{Z \rightarrow X}$ | 0.324 ± 0.197 | 0.003 ± 0.004 | 0.012 ± 0.014 | 0.657 ± 0.105 |
| $Y \rightarrow Z$ | -0.071 ± 0.078 | -0.005 ± 0.012 | -0.003 ± 0.023 | 0.005 ± 0.011 |
| $\mathbf{Z \rightarrow Y}$ | 0.101 ± 0.052 | -0.002 ± 0.006 | -0.003 ± 0.016 | 0.555 ± 0.109 |
| $AUC_{1 \rightarrow 2}$ | **1.00 (P<0.001)✗** | 0.27 (P=0.95)✓ | 0.31 (P=0.91) ✓ | 0.78 (P=0.02) ✓ |
| $AUC_{2 \rightarrow 1}$ | **1.00 (P<0.001)✗** | 0.2 (P=0.98)✓ | 0.31 (P=0.92) ✓ | 0.67 (P=0.09) ✓ |
| $AUC_{1 \rightarrow 3}$ | **0.98 (P<0.001)✗** | 0.41 (P=0.74)✓ | 0.61 (P=0.19) ✓ | 0.79 (P=0.02) ✓ |
| $AUC_{3 \rightarrow 1}$ | **0.93 (P<0.001)✓** | 0.25 (P=0.97) ✗ | **0.81 (P=0.01) ✓** | **1.00 (P<0.001) ✓** |
| $AUC_{2 \rightarrow 3}$ | 0.26 (P=0.97)✓ | 0.4 (P=0.76)✓ | 0.45 (P=0.63) ✓ | 0.46 (P=0.62) ✓ |
| $AUC_{3 \rightarrow 2}$ | 0.79 (P=0.02)✓ | 0.2(P=0.98) ✗ | 0.43 (P=0.69) ✗ | **1.00 (P<0.001) ✓** |
| NEURONS | | | | |
| COUPLED | | | | |
| $\mathbf{A \rightarrow B}$ | 0.267 ± 0.001 | 0.045 ± 0.014 | 0.028 ± 0.006 | 0.295 ± 0.012 |
| $A \leftarrow B$ | 0.055 ± 0.003 | 0.014 ± 0.009 | 0.026 ± 0.010 | 0.033 ± 0.012 |
| $AUC_{A \rightarrow B}$ | **1.00 (P=0.006)✓** | **1.00(P=0.006) ✓** | **1.00 (P=0.006) ✓** | **1.00 (P=0.006) ✓** |
| $AUC_{B \rightarrow A}$ | **1.00(P=0.006) ✗** | **1.00(P=0.006) ✗** | **1.00(P=0.006) ✗** | **1.00(P=0.006) ✗** |
| INDEPENDENT | | | | |
| $A \rightarrow B$ | -0.012 ± 0.001 | 0.001 ± 0.003 | -0.002 ± 0.008 | -0.006 ± 0.007 |
| $A \leftarrow B$ | -0.001 ± 0.001 | -0.001 ± 0.003 | -0.003 ± 0.005 | -0.002 ± 0.008 |

**Double Pendulum**. Our approach is the only one to recover the right causal direction from the sporadic data. The other baselines do not detect any significant correlation and thus no causal link between double pendulums. Despite having access to constant sampling data, multi-spatial CCM is also not able to detect the right data structure. We argue this is caused by the short length of time series window, and thus the low number and quality of delay embeddings that can be computed. In contrast, as our method shares the same model across all time windows, it represents more reliably the (hidden) state-space at any point in time. Importantly, the perfect reconstruction for Case 2 shows that we can distinguish confounding from correlation between time series. Indeed, when inferring causal directions between $X$ and $Y$, variable $Z$ is not used and thus *hidden*. Yet, our methods detects no causal relation between $X$ and $Y$. Figure 2a graphically presents the results of our method for the second case, where it is obvious that the only two convergent mappings are the ones corresponding to the true directions (solid blue and green lines), providing a strong signal for the right underlying causal mechanism.

**Neural activity** For the neuron activity data, we observe that our method delivers the largest effect size towards the true data generating model ($\mathcal{S}_c = 0.295$). Baselines methods relying on imputation do not provide any clear signal for a causal coupling (score 10 times lower). Multi-spatial CCM with the regularly sampled original data provides similar signal than our approach but dampened. Interestingly, we observe a small but significant correlation in the wrong direction ($A \leftarrow B$) suggesting a small coupling in this direction. An inspection of Figure 2b, however, will convince the reader that

the main causal effect is indeed from $A$ to $B$. This small correlation in the direction $A \leftarrow B$ is also observed in the fully observed data as shown in Figure 6 in Appendix B.

## 5  CONCLUSION AND FUTURE WORK

In this work, we propose a novel way to detect causal structure linking chaotic dynamical systems that are sporadically observed using reconstruction of underlying latent processes learnt with Neural-ODE models. We show that our method correctly detects the causal directions between temporal variables in a low and irregular sampling regime, when time series are observed in only short noncontiguous time windows and even in the case of hidden confounders, which are characteristics of real-world data. Despite the apparent limitation of our method to chaotic systems, it has been shown that CCM is broadly applicable in practice as many real dynamical systems are either chaotic or empirically allow Takens'-like embeddings. As our work builds upon CCM theoretically, we expect the range of application to be at least as large and leave the application to other real-world data for future work.

## ACKNOWLEDGEMENTS

YM is funded by the Research Council of KU Leuven through projects SymBioSys3 (C14/18/092); Federated cloud-based Artificial Intelligence-driven platform for liquid biopsy analyses (C3/20/100) and CELSA-HIDUCTION (CELSA/17/032). YM also acknowledges the FWO Elixir Belgium (I002819N) and Elixir Infrastructure (I002919N. This research received funding from the Flemish Government (AI Research Program). YM is affiliated to Leuven.AI - KU Leuven institute for AI, B-3000, Leuven, Belgium. YM received funding from VLAIO PM: Augmanting Therapeutic Effectiveness through Novel Analytics (HBC.2019.2528) and Industrial Project MaDeSMart (HBC.2018.2287) EU. This project has received funding from the European Union's Horizon 2020 research and innovation programme under the Marie Skłodowska-Curie grant agreement No. 956832. We also thank Nvidia for supporting this research by donating GPUs. EDB is funded by a SB grant from FWO.

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
