# OpenReview forum: "Latent Convergent Cross Mapping"
_ICLR.cc/2021/Conference — ICLR 2021 Poster_

### Official Review · AnonReviewer1 · 2020-10-15
**Extension of CCM to short time series using latent neural ODEs**

**Rating:** 6
**Confidence:** 4

**Review:**

### Summary
This paper studies short, chaotic time series and uses the Taken's theorem to discover the causality between two time series. The main challenge is that for short time series, the delay embedding is not possible. Thus, the authors propose to fit a latent neural ODE and theoretically argue that they can use the Neural ODE embeddings in place of the delay maps. The authors provide two sets of experiments, both on simulation data. Unfortunately, they never tested the algorithm on real data.

### Feedback
* The main confusion about this paper is the correlation function. $Corr(X, Y)$ is a symmetric function, likewise, the $S_{c X\to Y}$  defined on page 6 is symmetric too. How can we infer the direction of causality using a symmetric metric?
* The authors also do not describe why the $S_{c X\to Y}$ is an appropriate metric because it just finds the small sample bias in the estimation of $Corr(X, Y)$. Thus, it should be susceptible to the choice of the 100 data points. What is the expected value of this metric?
* The authors argue that their method works for short time series. But they use 100 data points as the initial point in Figure 2. Time series with length 100 is considered long in many applications.
* In Lemma 3.1, can you elaborate on how strong the following assumption is? How likely is it to hold for time series data in domains such as healthcare, finance, and natural sciences?
    > “If there exists one observation function $\alpha_H$ along with a valid couple $(k; \tau)$ (in the Takens’ embedding theorem sense) such that the map $\Phi(H(t))$ is injective,”
* The neural activity data is a point process. The ODE described in the paper is not an appropriate model for it. For example, associating the activity pattern with the missing values is inappropriate. Please use another ODE for this case.
* It would be nice to have a non-chaotic synthetic dataset and show the results of the proposed algorithm on it.
* The data generation in Section 4.3.2 uses missing totally at random. In reality, this seldom happens. The authors should use more realistic models for missing data.
* Please add markers to Figure 2.
* The authors are recommended to comment on how to apply their method to multivariate time series.
* Generally, sticking to the causality literature, I suggest the authors put the assumptions under which the results are causal in a clear statement. See for example the following paper for the statement of the assumptions:
    * Kennedy, E. H., Ma, Z., McHugh, M. D., & Small, D. S. (2015). Nonparametric methods for doubly robust estimation of continuous treatment effects. _Journal of the Royal Statistical Society. Series B: Statistical Methodology_, 79(4), 1229–1245.

-----
### Post-Response Update
After reading the authors' response and given the changes made in the paper, I increase my rating by one point.

---

> ### Author Response · Authors · 2020-11-14
> **Answer to reviewer 1**
>
> Dear Reviewer 1,
>
> Thank you for taking time to review our work ! Your comments greatly improved the paper. In particular, we clarified many important points as it appears that some sections might have led to some confusion about the foundations of our approach. Please find below a detailed answer to your comments.
> Thanks a lot !
>
> [About the pertinence of our score]
>
> We agree with you the notation previously used is confusing and we changed the paper accordingly. As you’ll see in section 3.1, what we called Corr(X,Y) is now replaced by Ccm(X,Y). Crucially, it does not stand for the correlation between process X and Y but rather for the correlation between the prediction of the latent / delay embeddings samples $\hat{\mathcal{M}}’_X$ from $\mathcal{M}’_Y$ and their true values $\mathcal{M}’_X$. Importantly, this score is not symmetric because the reconstruction of $\mathcal{M}’_X$ from $\mathcal{M}’_Y$ and the reconstruction of $\mathcal{M}’_Y$ from $\mathcal{M}’_X$ are different. Ccm(X,Y) compares the shadow manifold of X with its reconstruction while Ccm(Y,X) compares the shadow manifold of Y with its reconstruction.
> Furthermore, the score we choose has to reflect both requirements of CCM for a causal link, namely reconstruction signal (the strength of the correlation between the reconstruction of $\mathcal{M’}$ and its actual value) and the convergence of this correlation when the number of samples increases. As shown on Figure 2, causal coupling shows this clear convergent pattern of the reconstruction. In order to capture this pattern in a single score, for easier reporting, we check the difference between the correlation obtained with a low number of samples (we chose 100) and when all samples are used, as motivated in the original CCM paper.
> Importantly, 100 samples here corresponds to 100 samples of the latent process. As the latent process is sampled every 0.01 seconds, 100 samples of the latent process correspond to only one second of data, which corresponds to 4 samples on average (as described in Section 4.3 - data generation).
>
> [About the injectivity of the map between hidden process and delay embeddings]
>
> Assuming that the systems underlying healthcare, finance and natural sciences time series are chaotic in nature, which is probably the case, this will be valid. Indeed, most of the real world complex systems, even as simple as a non-conservative 3 body system, satisfy this condition.
> As stated in the paper, the Neural ODE architecture does not enforce injectivity of this map. However, we observe that this is the case in practice, due to regularization of the map from samples to hidden space. The derivation of a neural network architecture that would enforce it by design is left for future work.
>
> [About the ODE used for neural activity data]
>
> In our experiments we do not use the point process of  firing events of the neurons, but the average membrane potential of a group of neurons, simulating a continuous time measurement using an electrode in a tissue. The membrane potential is a continuously changing quantity and can be simulated by a set of ODE corresponding to ion channel dynamics. Actually our simulated data was created this way, and then subsampled the value to simulate missingness.
>
> [About non-chaotic datasets]
>
> Non-chaotic dynamical systems violate the assumptions of Takens theorem, therefore it is a limitation of the original CCM method, what our method inherits. Note that most of the real world complex systems satisfy this condition.
>
> [About missingness not at random]
>
> The data generated is indeed missing at random in our experiments. We are currently preparing another dataset where it is not the case. However, if the causal dependence between processes does not rely on the missingness pattern, we don’t expect it to impact our approach. As long as a reliable latent process is learnt, our causal detection mechanism should be valid. If the missingness not at random compromises the learning of a reliable latent process, another time series learning model is required, but the idea of our paper would remain.
>
> [About handling multivariate time series]
>
> As described in Section 3, our method applies by default to multivariate time series ($d_X$ can be larger than 1). In particular, both cases of double pendulums are multivariate time series with $d_X = 4$ (2 angles and 2 conjugate momenta).
>
> [About the assumptions for inferring causality in our work]
>
> We added more details about the notion of causality used in our work in Section 3 of the paper and detailed the connection with the concept of causality as defined by (Pearl 2009). In general, we require the general conditions of weak coupling (non synchrony) and chaotic time series.

---

> > ### Comment · AnonReviewer1 · 2020-11-20
> > **Few more questions**
> >
> > Thank you for the response.
> >
> > Can you provide references for the following statement? By far the most common approach for analysis of the time series in the following domains is non-chaotic modeling.
> >
> > > Assuming that the systems underlying healthcare, finance and natural sciences time series are chaotic in nature, **which is probably the case**, this will be valid.
> >
> > You have not responded to my question on the assumptions of Lemma 1.
> >
> > I am still unclear about the properties of the metric $Sc_{X\to Y}$. What is its expectation with respect to distribution of $X$?
> >
> > Can you explain in simple terms, why is there an asymmetry in the mutual information  $I(\mathcal{M}'_X, \hat{\mathcal{M}}'_X)\neq I(\mathcal{M}'_Y, \hat{\mathcal{M}}'_Y)$? This may help the audience better understand the idea behind your algorithm.

---

> > > ### Author Response · Authors · 2020-11-20
> > > **Additional answers and missingness not at random.**
> > >
> > > Dear Reviewer 1,
> > >
> > > Thank you for you feedback ! Please find below our answers to your comments.
> > >
> > > Best regards,
> > >
> > > The authors
> > >
> > > **About chaotic modeling applications**
> > >
> > > We agree with you, the most standard approach for those application areas is not chaotic modeling. Yet, due to the high prevalence of chaotic systems in nature [5], we argue that the fact that our method can harness this feature is an advantage. Several works have shown that addressing the chaotic nature of some systems was beneficial, with applications in finance and healthcare, among others [5,6,7,8]
> > >  Furthermore, our method extends CCM which has been already successfully used for a wide range of application areas like social media modeling [1], Healthcare [2], Climate science [3] or Economics [4] among others. As our method relies on the same assumptions as CCM, we thus expect it to apply for those applications as well.
> > >
> > > **About the assumptions of Lemma 3.1**
> > >
> > > If the system is chaotic, there always exists such a function $α$ along with a valid couple $(k;\tau)$  for an appropriately chosen latent process $H(t)$ (from Takens theorem [9]). So for applications in healthcare or finance, it depends if the underlying system can be described as chaotic. Examples include stock markets [6] or fMRI [2]. However, we have only empirical evidence that our model finds such an appropriate latent process $H(t)$.
> > >
> > > **About the score metric**
> > >
> > > $\mathbb{C}\mathrm{cm}$ refers to the correlation between the reconstruction of the delay embedding and its actual value. If both systems are independent, this empirical correlation has an expected value of 0. The score being the difference of this correlation computed with all samples and with only 100 samples, the expectation of the score for independent time series is thus 0. In the limit of infinite noiseless data samples and coupled systems, this score converges to $1-\delta$ where $\delta$ is the (low) correlation observed in the low sample regime, as it is then possible to reconstruct perfectly the attractor of the driving system from the attractor of the driven one (by Takens [9]).
> > >
> > > **About the asymmetry of the mutual information**
> > >
> > > Unidirectional causal link from X to Y results in a non-injective map from the delay embedding / latent space of Y to X. As an example, let’s take the function $M_x = M_y^2$. The reconstruction (with kNN) would then be (considering the large number of samples regime) :
> > > $\hat{M}_x = M_y^2$
> > > But
> > > $\hat{M}_y = 0$
> > > We then have $ 0 =  I(M_y;\hat{M}_y) <  I(M_x;\hat{M}_x)$
> > > We clarified this in the paper at the end of Section 3.1.
> > >
> > > **About missingness not at random**
> > >
> > > We added an Appendix section where we investigate the impact of the missingness not at random on our model (Appendix H). We used the following missingness pattern. If the absolute value of the angle of the first rod $\theta_1$ is larger than $\frac{\pi}{4}$, we sample the process with a probability two times larger than if the angle is smaller than $\frac{\pi}{4}$. We see that latent CCM can still infer the right causal direction with high confidence.
> > >
> > > **References :**
> > >
> > > [1] Social media modelling : Luo, C., Zheng, X., & Zeng, D. (2014, September). Causal inference in social media using convergent cross mapping. In 2014 IEEE Joint Intelligence and Security Informatics Conference.
> > >
> > > [2] Healthcare - FMRI : Wismüller, A., Wang, X., DSouza, A. M., & Nagarajan, M. B. (2014). A framework for exploring non-linear functional connectivity and causality in the human brain: mutual connectivity analysis (MCA) of resting-state functional MRI with convergent cross-mapping and non-metric clustering.
> > >
> > > [3] Climate Science : Brookshire, E. N. J., & Weaver, T. (2015). Long-term decline in grassland productivity driven by increasing dryness. Nature communications.
> > >
> > > [4] Economics : Liu, H., Lei, M., Zhang, N., & Du, G. (2019). The causal nexus between energy consumption, carbon emissions and economic growth: New evidence from China, India and G7 countries using convergent cross mapping.
> > >
> > > [5] In nature in general : Toker, D., Sommer, F. T., & D’Esposito, M. (2020). A simple method for detecting chaos in nature. Communications biology.
> > >
> > > [6] Kyrtsou, C.; M. Terraza (2003). "Is it possible to study chaotic and ARCH behaviour jointly? Application of a noisy Mackey-Glass equation with heteroskedastic errors to the Paris Stock Exchange returns series". Computational Economics.
> > >
> > > [7] Yang, H., Chen, Y., & Leonelli, F. (2016). CHARACTERIZATION AND MONITORING OF NONLINEAR. Healthcare Analytics: From Data to Knowledge to Healthcare Improvement, 59.
> > >
> > > [8] Denton, T. A., Diamond, G. A., Helfant, R. H., Khan, S., & Karagueuzian, H. (1990). Fascinating rhythm: a primer on chaos theory and its application to cardiology. American heart journal.
> > >
> > > [9] Floris Takens. Detecting strange attractors in turbulence. In Dynamical systems and turbulence, Warwick 1980, pp. 366–381. Springer, 1981.

---

### Official Review · AnonReviewer2 · 2020-10-23
**Relevant and novel paper for causal inference in dynamical systems, needs some clarification on critical aspects**

**Rating:** 7
**Confidence:** 4

**Review:**

The paper introduces Latent Convergent Cross Mapping (Latent CCM) that combines Neural ODEs with ideas from dynamical systems to detect causation between time series. The latent spaces learnt using Neural ODEs, and more specifically GRU-ODE-Bayes, are used to infer the direction of the causal links between two time series X[t] and Y[t].

Quality, clarity, originality and significance:
Pro: The paper is very well written, and the idea is interesting and quite novel. Inferring causation in time series is a significant problem in all areas involving dynamical systems.  Cons: As described below, there are a few critical conceptual aspects about the purpose of the method that need to be clarified, mainly related to the distinction between dynamical systems and time series.

One aspect that I find confusing is the lack of a clear distinction between the concepts of “dynamical systems” and “time series”. They seem to be used interchangeably, but there is a difference as time series are generated from dynamical systems. Thus, two different time series can still describe the same dynamical system, for example its different dimensions. The original paper for CCM states: “CCM uses Takens embedding to detect if two variables belong to the same dynamical system.”  The current paper seems to state that it can detect causation not only between two time series, but also between two different dynamical systems. I believe this is a critical aspect and should be clarified and justified. If it is between two time series, do they need to be generated by the same dynamical system?

The idea of using the latent space of a neural ODE to replace the delay embeddings in CCM is very nice, but I wonder if we can say that there is a one-on-one mapping between the time series from Neural ODE and the latent space? If this is not case, can we say that we can infer causation? Lemma 3.1 goes in this direction, but it depends on the existence of $\alpha_H$ and $(k, \tau)$. How easy/difficult is it to find $\Phi$ that is injective? This is not clear to me.

Different concepts (irregular, sporadic, incomplete, partially observed, missing, non-fully-observed) are used to define incomplete observations. In dynamical systems theory, “partial observations” often refers to the fact that only some of the dimensions of the dynamical system/attractor are observed. Having missing data is different: it might be that all dimensions are observed, but not at every timestep. Using a consistent terminology is needed, otherwise too many concepts to describe the same things (and potentially different things) make reading confusing because this can refer to missing/incomplete information on either the samples or the features/dimensions.

The plots in Fig. 1 are a bit too crowded, the missing data/points could be plotted with a lighter color, and indicate that the dots are the observed data in the caption (that was not clear to me right away). Going back to the discussion of time series vs. dynamical system, $X[t]$ and $Y[t]$ in Fig. 1 seem to be from the same dynamical system, is it the same Lorenz attractor with the same parameters?

The paper states that it doesn’t use delay embedding, however by using segments in a way it is doing something very similar to delay embedding because it uses information about the past trajectory of a point as one single input to identify the latent space. This could also potentially explain why using different segments helps to find the latent state space. Could the authors extend on this connection? Do the authors believe that the method would work if only the individual points would be given as input, and not segments?

Sect. 3: are $d_X$ and $d_Y$ different? I don’t see clearly the connection with the do operation, as it is not mentioned afterwards? Would be good to clarify what $t_X$ and $t_Y$ are, it is not very intuitive.

Sect. 3.3: How could one check or guarantee that the “hidden representation is a complete representation of the state-space”? It would be great to give some details on GRU-ODE-Bayes as this is the method used.

Lemma 3.1.: Should it be $dH(t)/dt$? Before, the subscript for $\Phi$ was $\phi$ not $g(\phi)$. What is the difference?

Sect. 4.2.: What are the “both aspects”? Why 100 samples for Corr_0, and could the authors explain a bit more the reasoning behind the choice of the score? Intuitively, I would have thought a score that is a percentage between the two correlations would be easier to interpret, but maybe I am wrong?

Sect. 4.3.1: Which is the quadratic potential? What happens for noisy observations? Have the authors tried adding noise and see if the causal inference still works?

Fig. 3: should be Y -> X instead of X-> Y

Sect. 4.5: What do the authors mean by folds here, is there a cross validation step? In the Neural activity paragraph, it should be A and B instead of X and Y.

Table 1: Could the authors name the methods and describe them briefly? The acronym MVGP is only explained in the appendix. In the first case, for Latent CCM, the score is significantly smaller than 1, and very close to zero. Is there a bound/threshold under which we cannot say that we infer causality?

What is the connection between this paper and the work presented in Rubanova et al, 2019: Latent ODEs from irregularly-sampled time series?

-----------------------------------
Rebuttal: Thank you to the authors for considering the comments and for the changes. I am happy with the response, and have increased the score accordingly.

Fig. 5 could be changed similarly to Fig. 1 for visual clarity. The authors mention in the response that both GRU-ODE-Bayes and Rubanova et al. 2019 can be used interchangeably, but there is no reference to the latter.

---

> ### Author Response · Authors · 2020-11-15
> **Answer to Reviewer 2 [Part 1/2]**
>
> Dear Reviewer 2,
>
> First of all thank you for this comprehensive review of our work and for your suggestions that greatly improved the paper. We amended the manuscript with your suggestions and it’s now available on Openreview. Importantly, we clarify the distinction between time series and dynamical systems and give more details about our notion of causality in the beginning of Section 3 and in the whole paper in general. Typos in the arrow on Figure 3 and in Section 4.5 have also been fixed.  Please find a more detailed answer to your comments below. The answers have been split in 2 distinct OpenReview comments. Apologies for the slightly going over the words limit but we wanted to provide comprehensive answers.
>
> Thanks a lot !
>
>
> [About the distinction between time series and dynamical systems]
> We totally agree with you and we updated the paper to reflect this distinction between time series and dynamical system. Here, we refer to the dynamical system of a variable X, as the smallest dynamical system (or system of ODE equations) defining the dynamics of X (see Section 3 in the revised version of the paper). Our notion of causality is then that if X causes Y then X has to be part of the dynamical system of Y.
> As an example, let’s take the following system of ODEs representing variables X and Y.
>
>    $ \frac{dX(t)}{dt} = f(X(t)) \quad (1) $
>     $\frac{dY(t)}{dt} = g(X(t)) + h(Y(t)) \quad (2) $
>
> The dynamical system of X is (1). While the dynamical system of Y is (1) + (2) as (1) is needed to describe the dynamics of Y. Our notion of causality is then inline with the Pearl definition of causality: (P(Y | do(X)) is not equal to P(Y)). It is clear from (1) and (2) that setting X(t) (intervention on X(t)) would impact Y(t).
>
> [About the requirements for Lemma 3.1]
> Indeed, in Lemma 3.1, we give the conditions for a one-on-one mapping between the latent process and the delay embedding. As stated in the paper, the Neural ODE architecture does not enforce injectivity of this map. However, in practice, due to regularization of the map from samples to hidden space, we observe that this is the case. The design of new architectures enforcing injectivity of the map between latent process and delay embedding is very interesting and is left for future work.
>
> [About using a consistent wording for the concept of sporadic time series]
> The case we consider here is both partially observed (not all dimensions of the dynamical system are observed) and with missing data. In the case of the neural data, we only observe the average potential of a population of neurons (partially observed). By sporadic, we mean both missing data and also the non-constant sampling of the observations. We thus eliminated non-fully-observed and irregular from the text to avoid confusion.
>
> [About using individual points as inputs]
> We don’t use explicit delay embeddings but we indeed need to observe some short segments. With individual points, the temporal/dynamic information of the data would be lost and only instantaneous coupling between time series could theoretically be recovered. Instead, we allow for interaction between time series with arbitrary delay in the coupling. This requires temporal information.
> As we consider that the set of short time series comes from the same dynamical system, we can learn the dynamics ($f_{\theta}$ in Equation 1 in the paper) jointly over the whole set of short time series, which is a strength of our method over classical approaches. Of course, would the time series become too short, our method would start having issues to learn the correct dynamics as well. Eventually, when the time series becomes too short, the long term dependencies of the process are lost (this is valid for any method).
> Overall, sufficient information/samples are needed to learn a faithful representation of the data generating process. Yet, the required information is lower than for classical CCM as shown in Results Section 4.5.
>
> [About notations]
> dX and dY can indeed be different. We dropped the tX and tY in the paper as it was not used later. More details about the link with the do operation are now given in Section 3 of the paper and in the first comment of this answer.
>
> [About Lemma 3.1]
> It is indeed $\frac{dH(t)}{dt}$. We updated the paper accordingly. $g(\phi)$ is used here because we integrate the process $H(t)$ over time and then apply the function $g(\cdot)$ that will map it to the space $\mathcal{X}$.

---

> ### Author Response · Authors · 2020-11-15
> **Answer to Reviewer 2 [Part 2/2]**
>
> [About the score in section 4.2]
> The score we choose has to reflect both requirements of CCM for a causal link, namely reconstructing the signal (the strength of the correlation between the reconstruction of $\mathcal{M’}$ and its actual value) and the convergence of this correlation when the number of samples increases. As shown on Figure 2, causal coupling shows this clear convergent pattern of the reconstruction. In order to capture this pattern in a single score, for easier reporting, we check the difference between the correlation obtained with a low number of samples (we chose 100 latent samples) and when all samples are used, as motivated in the original CCM paper.
>
> [About section 4.3.1 - (Now 4.3 - Description)]
> The term we refer to in Section 4.3 is $(\theta_1^Y-\theta_1^X)$. This corresponds to a quadratic term in the potential of the system.
> Indeed, the imposed quadratic potential is $U(\theta_1^Y) = c_{X,Y} (\theta_1^Y-\theta_1^X)^2$. Please note that the potential is unphysical, as it is not symmetric in general ( $c_{X,Y} \neq c_{Y,X}$ ).
> The force is the gradient of the potential:
> $\frac{dU(\theta_1^Y) }{d\theta_1^Y} = - 2  c_{X,Y} (\theta_1^Y-\theta_1^X)$. This corresponds to a hypothetical force imposed by a stretched out spring with spring constant $c_{X,Y}$ on system Y, assuming system X is fixed in space. It can be realized in practice with a controller.
> Both cases considered have additive noise as described in Section 4.3 Data Generation.
>
> [About the meaning of folds]
> We agree that fold is misleading and we here mean 5 repeats. Each repeat is from the same dynamical system structure but with different initial conditions. We further give more details about validation in the new Appendix G.
>
> [About Table 1]
> We added a reference to MVGP in section 4.1 about baselines. Regarding the first case, the high level of noise reduces the strength of the correlation and makes it less clear to detect causal direction. However, despite the small value, it is still significant. Note that, as suggested by reviewer 3, we also included comparison with 2 other baselines as described in Appendix F.
>
> [About the similarity with Rubanova et al., 2019]
> GRU-ODE-Bayes and the method proposed in Rubanova et al. are indeed very similar and could be used interchangeably to learn the dynamics of the processes. Both works were published concomitantly. In this paper, we show that a Neural ODE model (GRU-ODE-Bayes or Rubanova et al.) can be used to learn causal dependencies.

---

> ### Author Response · Authors · 2020-11-22
> **Increased readability of Figure 1**
>
> Dear Reviewer 2,
>
> As you suggested, we updated Figure 1 of the manuscript to be more readable.
>
> We are at your disposal to answer any other question you might have about our work.
>
> Best Regards,
>
> The authors.

---

### Official Review · AnonReviewer3 · 2020-10-28
**Good work, but I have some concerns in terms of experiments and complexity of the method.**

**Rating:** 6
**Confidence:** 4

**Review:**

This work proposes latent CCM, a causal discovery method for short, noisy time series with missing values. The method checks whether there exists CCM between latent processes of the time series without computing delay embeddings. Empirical results show the proposed method is more accurate in finding the right causal direction in various datasets.

One of my major concerns is: Is it necessary to learn continuous time latent representation? As time series are not continuous. It would not be useful to go beyond the granularity of the original time series. And solving the problem of missing value does not necessarily require “continuous time latent representation”. Also in the Double Pendulum results, the authors claim the improvement over the multispatial CCM is because the proposed model shares the same parameters across time windows, not because of the continuity of the model. So, it would be interesting to see an ablation study with a model which is based on GRU but without Neural ODEs.

In 4.3.2, the authors mentioned, “We used 80% of available windows for training and used the remaining 20% for hyperparameter tuning”. It would be better to specify what criterion is used for model selection. In addition, one good practice in this kind of causal direction discovery is to anonymize the two time series in training and validation (avoid using the ground truth in model selection), it is not clear if the authors followed this routine.

It would be interesting to report the computational complexity (time and space) of the proposed method. In addition, it would be appreciated if the authors can also report the runtime comparison between the algorithms.

It would also be better if the choice of baselines can be more comprehensive. For example, the authors can try to include algorithms beyond the CCM family such as PCMCI [1] and VARLINGAM [2].

In terms of writing and presentation, It would be better to make the content self-contained and make the use of terminologies consistent. For example, the authors did not explain terms like "synchrony" and "confounding case" etc.


[1] Runge, Jakob, Peer Nowack, Marlene Kretschmer, Seth Flaxman, and Dino Sejdinovic. "Detecting and quantifying causal associations in large nonlinear time series datasets." Science Advances 5, no. 11 (2019): eaau4996.

[2] Hyvärinen, Aapo, Kun Zhang, Shohei Shimizu, and Patrik O. Hoyer. "Estimation of a structural vector autoregression model using non-gaussianity." Journal of Machine Learning Research 11, no. 5 (2010).

---

> ### Author Response · Authors · 2020-11-14
> **Answer to reviewer 3**
>
> Dear Reviewer 3,
>
> First of all, thank you for providing a comprehensive review of our paper. We addressed your comments in the revised version of our manuscript that has been uploaded to Openreview. In particular, the concern about the necessity of the continuity of the latent process has been discussed in section 3.3, more details about the cross-validation have been added in section 4.3.2 and we added an Appendix section (Appendix G) for a more in depth discussion. We compared our method against the baselines you suggested (PCMCI and VARLinGAM) in the baselines section (4.1) and added an Appendix section with detailed results and discussion (Appendix F). We also improved the manuscript by making it more self-contained, providing definitions for terms like synchrony, confounding, attractor and dynamical system. We are currently working on a report of the complexity of the method as well as a proper ablation study of the Neural ODE. Finally,  please find below more detailed answers to your comments.
>
> Thanks again,
>
> The authors.
>
> [About the necessity of a continuous latent process]
> The main advantage of modeling the latent process as a continuous process is that it gives more coverage of the hidden space (i.e. we will have access to more distinct points on the underlying manifold $\mathcal{M}’$).
> With a regular GRU, the hidden process in between observations would be constant, leading to fewer distinct points in the latent space. As the method relies on the ability to recover an attractor from another, it is crucial to have a good coverage of it. A continuous representation will help in the characterization of the attractors. What is more, the sampling rate applied to the time series is much lower than the intrinsic frequency of the process, such that the dynamics of the hidden process in between observation is important. As an illustration, we refer you to Figure 5 of the Appendix where it is clear that the value of the latent process in between observations will give significant information about the manifold of interest $\mathcal{M}’$. With a regular GRU, the hidden process in between observations would be constant.
> To provide more tangible proof of this above claim, we are currently preparing an ablation study replacing the ODE with a regular GRU that we hope to incorporate in the revised manuscript very soon. In the meantime, we also refer to the paper of De Brouwer et al. where such an ablation study had been performed and where it was shown that the ODE was beneficial when the data was sampled from a continuous process, which is the case in our simulations.
>
> [About cross-validation]
> For the cross-validation, MSE on future samples was used (see Appendix G). The learning of the dynamics of both time series is done *separately*, leading to 2 different Neural ODE models for each time series. On the other hand, the causal direction detector is not trained or tuned to the data (it checks if there exists a map from one hidden space to another, treating the two time series symmetrically - hence anonymized). There is thus no risk of contamination from one time series to the other. No information or label about the causal direction is used at any point in the process.
>
>
> [About extending the set of baselines]
> We included both suggested baselines as comparison (Section 4.1 - Baselines and Appendix F). Please note that those methods could not handle sporadic time series so we instead used a less challenging version of the data where the sampling is constant and without missingness across dimensions. As shown in table 6, PCMCI finds the underlying causal links 40% of the time for case 1 and fails to uncover the right links for case 2. VarLinGAM does better for case 1 (60%) but fails for case 2. One reason for poor performance of those methods can be the fact that they rely on time lags and check the dependence between time series for each time lag, rather than considering a full notion of causality between temporal variables as we propose.
>
> [About making the manuscript more self-contained]
> We added a definition for synchrony and for confounding case in section 2. We also added a definition of attractor (section 3.1) and dynamical system (section 3) to be more self-contained. Please find below our definitions:
> *(Generalized) Synchrony : in the case of a dynamical system X driving Y, generalized synchrony occurs when $Y(t) = \phi(X(t))$ for some function $\phi$. Both systems are then equal up to some map $\phi$ and the attractors are homeomorphic to each other. This occurs when the coupling is too strong.
> *Confounding case : When two variables are causally driven by a third one. In general, X confounds the relation between Y and Z if X causes both Y and Z.
> *Dynamical system : we refer to the dynamical system of a time varying variable X as the smallest dynamical system that fully describes the dynamics of X.
> *Attractor : manifold toward which the state of a chaotic dynamical system tends to evolve.

---

> ### Author Response · Authors · 2020-11-22
> **Complementary answers**
>
> Dear Reviewer 3,
>
> Thanks again for reviewing our work. We wanted to provide complementary information on top of our last answer, regarding the comparison between Neural-ODE approaches and RNN for the creation of the embedding in our approach and about the complexity of our method and baselines.
>
>
> **Regarding the usage of a Neural-ODE method for learning the latent generative process of the data**
>
> As Equation 3 in the paper suggests, we consider our observations are generated from a continuous latent process $H(t)$. Different techniques could be used to infer this process $H(t)$ in our latent CCM approach. Among those techniques, neural-ODE models such as [1] or [2] embody the assumptions of Equation 3 and are thus a natural choice for the inference of informative latent vectors. Another choice could be to learn those dynamics with a non-continuous recurrent neural network approach. In the newly added Appendix section I, we present results from latent CCM using a GRU for the inference of the latent process where we observe that the approach leads to the inference of an incorrect causal graph. Furthermore, we want to reiterate several theoretical properties of neural-ODEs methods that present an advantage over standard RNNs in our causal inference approach. Neural-ODEs allow to have a denser coverage of the attractor we want to reconstruct. This feature is further strengthened by the fact that different integrators can be used to recover the latent process, therefore allowing to tune the resolution of the learnt latent process. In the case of physical systems, symplectic integrator can also be used, to ensure conservation of energy and more accurate learning of the dynamics [3]. Finally as shown in [1], a continuity prior is beneficial in the low number of samples regime.
>
> **Regarding the complexity of our method and of the baselines**
>
> We added a section in the Appendix specifying the complexity of our method and of all compared baselines (Appendix J).
>
> **References :**
>
> [1] De Brouwer, E., Simm, J., Arany, A., & Moreau, Y. (2019). GRU-ODE-Bayes: Continuous modeling of sporadically-observed time series. In Advances in Neural Information Processing Systems (pp. 7379-7390).
>
> [2] Rubanova, Y., Chen, R. T., & Duvenaud, D. K. (2019). Latent ordinary differential equations for irregularly-sampled time series. In Advances in Neural Information Processing Systems (pp. 5320-5330).
>
> [3] Zhong, Y. D., Dey, B., & Chakraborty, A. (2019). Symplectic ode-net: Learning hamiltonian dynamics with control. arXiv preprint arXiv:1909.12077.

---

### Official Review · AnonReviewer4 · 2020-10-28
**Review of "Latent convergent cross mapping"**

**Rating:** 6
**Confidence:** 2

**Review:**

##################################################

Summary:
The paper provides an interesting extension to the convergent cross mapping methods. It proposes a latent process model to learn the directions of causal mechanisms from time series observed at irregular intervals. The method is also developed to be suitable for the situation where the time series are observed at short segments.

##################################################

Pros:
The Neural ODE method is interesting and novel. It solves the problem of having irregular time series in the applications where the dynamic systems have been extensively studied and represented by ODEs.

##################################################

Cons:
1. The causality established by looking at H_x and H_y is not fully described vigorously. There has been much recent work in the traditional causal inference literature where latent variables are involved. However, the theoretical results are generally difficult to establish. In the context of this paper, more justification is needed to convince readers that you can replace X and Y with their (estimated) latent process counterparts. Do you need to assume there is enough information to exactly reconstruct the latent process? Will you run into model misspecification problem of the latent process as both time series are short?

2. In the conclusion it is claimed the method can detect causal directions even in the case of hidden confounders. I do not see from the main text where this property is discussed. Case 2 of the simulation seems to be correspondong to this statement, but is Z[t] hidden in the simulation? If so it should be made clearer. But in general, I do not quite see how the lack of casual arrows from X to Y can be recovered when there is a hidden confounder Z. Maybe it is possible in some particular cases. Can you clarify?

---

> ### Author Response · Authors · 2020-11-14
> **Answers to reviewer 4 - comment 1/2.**
>
> Dear Reviewer 4,
>
>
> Thank you for taking the time to review our work. We addressed your comments by amending our paper accordingly (see new version). We addressed most of your first comment at the beginning of the “Method” section (page 3). We incorporated your second comment in section 4.3.2 (we clarify the pairwise causal inference) and in section 4.5 - Double Pendulum ( we clarify the hidden confounder setup). We also give more detailed answers to your insightful comments below.
>
> *Comment 1. About our description of the notion of causality.*
>
> We refer to the dynamical system of a time varying variable X as the smallest dynamical system that fully describes (without external variables) the dynamics of X. As an example, let’s take the following system of ODEs representing variables X and Y.
>
>
> $   \frac{dX(t)}{dt} = f(X(t)) \,(1) $
> $    \frac{dY(t)}{dt} = g(X(t)) + h(Y(t))\,(2)$
>
> The dynamical system of X is (1). While the dynamical system of Y is (1) + (2) as (1) is needed to describe the dynamics of Y.
> Our notion of causality is then that if X causes Y then X has to be part of the dynamical system of Y. This is inline with the Pearl definition of causality (P(Y | do(x)) is not equal to P(Y)). It is clear from (1) and (2) that setting x(t) (intervention on X(t)) would impact Y(t).
> Therefore the attractor of the dynamical system of Y will contain information about the attractor of the dynamical system of X. And it should be possible to recover the attractor of the dynamical system of X from the attractor of the dynamical system of Y.
> However, getting a representation of the attractor of a dynamical system based on one of its time series is not straightforward. One way to do it is to use Takens theorem (delay embeddings). Indeed, takens theorem states that delay embeddings of a time series is a diffeomorphism of the attractor of its dynamical system. The problem is that computing delay embeddings is not always possible (when time series are short and sporadic).
> What we show empirically and with theoretical foundations in this paper is that the learnt hidden process is also a valid representation (an embedding) of the attractor (just like the delay embeddings) and it gives another way of representing the attractor of the underlying dynamical system.
>
> [About the need to have enough information to learn the latent process] As you point out, we usually require sufficient information (or samples) to learn a latent process that predicts values of X with sufficient accuracy over the time scale defined by the delay embedding (Takens) theorem, which depends on the number of intrinsic dimensions and the auto-correlation of the system. The result section shows that the number of samples required is actually lower than the one required for classical convergent cross mapping. Additionally, Figure 5 in the Appendix shows that despite a sparse sampling, the model is able to recover the true dynamics of the system accurately.
> Importantly, let’s also note that given a time series X[t], there will be many processes that satisfy Equation (1) in the paper, so it suffices to learn one that represents X correctly.
>
> [About the length of the time series] As we consider that the set of short time series comes from the same dynamical system, we can learn the dynamics ($f_{\theta}$ in Equation 1 in the paper) jointly over the whole set of short time series, which is a strength of our method over classical approaches. Of course, would the time series become too short, our method would start having issues to learn the correct dynamics as well. Eventually, when the time series becomes too short, the long term dependencies of the process are lost (this is valid for any method).
> Overall, sufficient information/samples are needed to learn a faithful representation of the data generating process. Yet, the required information is lower than for classical CCM.
>
> Please see the following below answer for your second comment.

---

> ### Author Response · Authors · 2020-11-14
> **Answers to reviewer 4 - comment 2/2.**
>
> *Comment 2 - Clarification about the ability to deal with the hidden confounding case.*
>
> Indeed, it corresponds to case 2. Importantly, and as we now made clearer in the paper, causality between time series is assessed in a *pairwise* fashion. In case 2 of the simulations, causality between X and Y is assessed without taking Z into account, which then becomes a hidden common cause of X and Y. Yet, the method is able to detect that there is no direct causal relation between X and Y.
>
> However, when checking pairwise causality between X and Z (Z is then therefore observed here and Y is hidden), our method uncovers the correct causal link between Z and X. Similarly, it detects the right causal link between Y and Z.
> This is possible because of the CCM rationale on chaotic dynamical systems. Indeed the dynamical system of X does not contain Y (the converse is also true). But the dynamical system of X contains Z and the dynamical system of Y contains Z as well. As an example, let’s consider the general dynamical system corresponding to the causal graph : X <- Z -> Y .
>
> $\frac{dX}{dt} = f(X) + g(Z)$
> $\frac{dY}{dt} = h(Y) + j(Z)$
> $\frac{dZ}{dt} = m(Z)$
>
> The dynamical system of $X$ writes :
>
> $\frac{dX}{dt} = f(X) + g(Z)$
> $\frac{dZ}{dt} = m(Z)$
>
> The one of $Y$ writes :
>
> $\frac{dY}{dt} = h(Y) + j(Z)$
> $\frac{dZ}{dt} = m(Z)$
>
> And the one of $Z$ writes :
>
> $\frac{dZ}{dt} = m(Z)$
>
> We clearly see that only the dynamical system of $X$ and $Y$ contain the one of $Z$, and hence only those causal relations are recovered.
> As it is well established in causality literature, no method can detect hidden confounders from observational data alone, without imposing some assumptions on the system. In the case of CCM, existence of an attractor (therefore applicability of Takens theorem) is such a condition. Note that most of the real world complex systems, even as simple as a non-conservative 3 body system, satisfy this condition.

---

### Decision · Program_Chairs · 2021-01-07
**Final Decision**

**Decision:**

Accept (Poster)

**Comment:**

This paper is concerned with finding causal relations from temporal processes and extends the Convergent Cross Mapping (CCM) method.  It focuses on finding information of chaotic dynamical systems from short, noisy and sporadic time series, and the idea of using the latent space of neural ODEs to replace the delay embeddings in CCM seems interesting. All reviewers like the idea. Please try to make the paper more self-contained and provide some of the justifications suggested by the reviewers.